# Preventive Triple Gene Therapy Reduces the Negative Consequences of Ischemia-Induced Brain Injury after Modelling Stroke in a Rat

**DOI:** 10.3390/ijms21186858

**Published:** 2020-09-18

**Authors:** Vage Markosyan, Zufar Safiullov, Andrei Izmailov, Filip Fadeev, Mikhail Sokolov, Maksim Kuznetsov, Dmitry Trofimov, Evgeny Kim, Grayr Kundakchyan, Airat Gibadullin, Ilnur Salafutdinov, Leniz Nurullin, Farid Bashirov, Rustem Islamov

**Affiliations:** 1Department of Medical Biology and Genetics, Kazan State Medical University, 420012 Kazan, Russia; vage.markosyan@gmail.com (V.M.); redblackwhite@mail.ru (Z.S.); gostev.andrei@gmail.com (A.I.); philip.fadeyev@gmail.com (F.F.); supermihon@yandex.ru (M.S.); qmaxksmu@yandex.ru (M.K.); t.dima19961996@gmail.com (D.T.); profzh@yandex.ru (E.K.); halsoulo@gmail.com (A.G.); faridbashirov@yandex.ru (F.B.); 2Institute of Fundamental Medicine and Biology, Kazan [Volga Region] Federal University, 420008 Kazan, Russia; grayr23@gmail.com (G.K.); sal.ilnur@gmail.com (I.S.); 3Kazan Institute of Biochemistry and Biophysics, Federal Research Center of Kazan Scientific Center of Russian Academy of Sciences, 119991 Kazan, Russia; leniz2001@mail.ru

**Keywords:** stroke, preventive gene therapy, adenoviral vector, VEGF, GDNF, NCAM, human umbilical cord blood mononuclear cells

## Abstract

Currently, the main fundamental and clinical interest for stroke therapy is focused on developing a neuroprotective treatment of a penumbra region within the therapeutic window. The development of treatments for ischemic stroke in at-risk patients is of particular interest. Preventive gene therapy may significantly reduce the negative consequences of ischemia-induced brain injury. In the present study, we suggest the approach of preventive gene therapy for stroke. Adenoviral vectors carrying genes encoding vascular endothelial growth factor (VEGF), glial cell-derived neurotrophic factor (GDNF) and neural cell adhesion molecule (NCAM) or gene engineered umbilical cord blood mononuclear cells (UCB-MC) overexpressing recombinant VEGF, GDNF, and NCAM were intrathecally injected before distal occlusion of the middle cerebral artery in rats. Post-ischemic brain recovery was investigated 21 days after stroke modelling. Morphometric and immunofluorescent analysis revealed a reduction of infarction volume accompanied with a lower number of apoptotic cells and decreased expression of Hsp70 in the peri-infarct region in gene-treated animals. The lower immunopositive areas for astrocytes and microglial cells markers, higher number of oligodendrocytes and increased expression of synaptic proteins suggest the inhibition of astrogliosis, supporting the corresponding myelination and functional recovery of neurons in animals receiving preventive gene therapy. In this study, for the first time, we provide evidence of the beneficial effects of preventive triple gene therapy by an adenoviral- or UCB-MC-mediated intrathecal simultaneous delivery combination of *vegf165*, *gdnf*, and *ncam1* on the preservation and recovery of the brain in rats with subsequent modelling of stroke.

## 1. Introduction

The current options for ischemic stroke treatment are extremely limited and are aimed at restoring blood flow in the ischemic area by intravenous infusion of recombinant tissue plasminogen activator and/or physical removal of the clots [1]. To date, the main fundamental and clinical interest is focused on developing a neuroprotective treatment of the penumbra region within the therapeutic window. The strategy of cell-, gene-, and gene-cell therapy for neuroprotection in stroke treatment has been proven by numerous experiments in animal models [2,3,4]. Besides brain-specific cell types, umbilical cord blood (UCB) is widely used for neuroprotection in the central nervous system (CNS) for different pathological conditions [5]. UCB cells are considered a valuable source of stem cells, growth and neurotrophic factors for cell therapy. The mononuclear fraction of UCB contains populations of different immature cells that are capable of differentiating into many cell types [6] and, thus, represent an alternative to embryonic stem cells for transplantation to patients with post-ischemic, post-traumatic and degenerative diseases [7,8]. To date, the following have been discovered in UCB: Hematopoietic stem cells (HSCs), endothelial progenitor cells, mesenchymal stem cells (MSCs), unrestricted somatic stem cells (USSCs), and side population cells (SP) [9,10,11,12].

Due to the immaturity of the immune system of a new-born, the use of UCB cells for cell therapy does not require matching of genes relating to HLA (Human Leucocyte Antigens) human tissue compatibility, as evidenced by the absence of an acute or chronic form of the disease “graft-versus-host” (graft versus host disease) [13,14]. In addition, with UCB cell transplantation, tumor transformation of cells in the recipient’s body is practically prevented [15].

Another attractive reason for using UCB cells for cell therapy is their ability to produce various biologically active molecules, such as proteins which are antioxidant, angiogenic, neurotrophic, and growth factors [16,17,18,19,20]. Thus, transplantation of UCB cells can be aimed at replacing dead cells and at preventing the further death of surviving cells due to secreted biologically active molecules. Enhancement of the positive effects of UCB cells on tissue regeneration after their genetic modification is a relatively new and promising gene-cell approach in cell therapy to stimulate post-traumatic or post-ischemic brain injury [21,22]. Gene-modified UCB cells may provide addressed delivery of therapeutic genes and supply the expression of the recombinant molecules at the site of regeneration.

In our previous studies, we showed the positive effect of gene-modified umbilical cord blood mononuclear cells (UCB-MC), simultaneously producing three recombinant molecules—vascular endothelial growth factor (VEGF), glial cell-derived neurotrophic factor (GDNF) and neural cell adhesion molecule (NCAM)—in animal models of amyotrophic lateral sclerosis [23], spinal cord injury [24], and stroke [25]. The rationale of using a combination of two neurotrophic factors with cell adhesion molecules is based on the well-known neuroprotective effects of VEGF and GDNF [26,27], with the expression of NCAM increasing the homing and survivability of UCB-MC at the brain injury site [28] supporting local production of the therapeutic molecules. In the model of middle cerebral artery occlusion (MCAO) in rats, we demonstrated that intrathecal injection of genetically-engineered UCB-MC over-expressing VEGF, GDNF, and NCAM, four hours after MCAO results in a reduction of infarct volume, the positive reaction of neuroglial cells and an increase in synaptic protein expression. Thus, ex vivo gene modification may enhance the naïve neuroprotective properties of UCB-MC.

The development of treatment under the threat of stroke is of particular interest. Preventive therapy may highly reduce the consequences of a stroke-induced brain injury. In the present study, we suggest the approach of preventive cell-mediated gene therapy for stroke. The efficacy of gene-engineered UCB-MC overexpressing recombinant molecules-stimulants of neuroregeneration VEGF, GDNF, and NCAM, administered intrathecally 3 days before MCAO in rats, was investigated using morphometric and immunofluorescent methods.

## 2. Results

### 2.1. Molecular Analysis of Gene-Modified UCBC

The efficiency of UCB-MC transduction by Ad5-GFP adenoviral vector was confirmed after 72 h of UCB-MC+Ad5-GFP cultivation using fluorescent microscopy. In the cytoplasm of UCB-MC+Ad5-GFP, a specific green glow was detected (Figure 1A). By the flow cytometry method, it was established that the percentage of GFP-positive human UCB-MC at multiplicity of infection (MOI) equal to 10 reaches 29.5% (Figure 1B).

Evaluation of mRNA levels of transgenes (*vegf165*, *gdnf*, and *ncam1*) in genetically-modified UCB-MC was performed by RT-PCR, 72 h after incubation of UCB-MC+Ad5-VEGF-GDNF-NCAM. The molecular analysis revealed that the content of mRNA *vegf165* increased 141.8 ± 8.24 times, *gdnf* mRNA 167.51 ± 6.85 times and *ncam1* mRNA 122.9 ± 13.5 times compared with intact (naïve) UCB-MC (Figure 1C).

Multiplex analysis of cytokines, chemokines and growth factors in supernatants harvested after the cultivation of naïve UCB-MC identified a wide range of pro- and anti-inflammatory cytokines, chemokines and growth factors (Figure 1D). However, the concentrations of Flt-3L, IL12-p70, IL-15, IL-17a, IL-9, and TNFb were lower than the detection level. Genetic modification of human UCB-MC by recombinant adenovirus Ad5-GFP does not affect the secretory profile of modified cells in relation to the investigated factors when compared to the naïve cells. UCB-MC simultaneously transduced with Ad5-VEGF, Ad5-GDNF, and Ad5-NCAM also preserves the qualitative and quantitative profile of expression of the investigated factors. However, as expected, in comparison with the naïve and UCB-MC+Ad5-GFP, we observed a 200-fold increase of VEGF level. The presented results correlate with PCR-RT data above and confirm the efficiency of UCB-MC transduction and its ability to synthesize and secrete recombinant molecules, as with the example of VEGF. In addition, the results support that adenoviral vector does not affect the production of the studied biologically active molecules by human UCB-MC.

### 2.2. Morphometric Analysis of Infarct Area

Three weeks after ischemic stroke modelling, cerebral infarction volume analysis revealed an infarct zone located in the parietal lobe (parietal cortex, area 1 (Par1), which corresponds to the site of MCAO (Figure 2A,B). Morphometric analysis of the brain cortex infarct cavities volume showed the differences between therapeutic and control groups (Figure 2C,D). The infarct cavities volume was significantly less in the Ad5-VEGF-GDNF-NCAM (0.177 [0.155; 0.197]) and UCB-MC+Ad5-VEGF-GDNF-NCAM (0.070 [0.014; 0.245]) groups when compared with the control saline (0.607 [0.568; 0.759]) and Ad5-GFP (0.817 [0.754; 0.865]) groups (*p* < 0.05). In the UCB-MC+Ad5-GFP (0.249 [0.119; 0.305]) group, the infarction volume did not differ from the gene-treated groups and was lower when compared with the saline group (*p* < 0.05) (Figure 2E).

### 2.3. Immunofluorescent Study of Brain

Comparative analysis of the molecular and cellular changes in the peri-infarct zone (Figure 2C) of experimental rats’ brains revealed different patterns in the expression of cellular stress, apoptosis, and synaptic proteins and in the reorganization of neuroglial cells in relation to data obtained from intact animals.

#### 2.3.1. Cellular Stress and Apoptosis Proteins 

The expression of heat shock protein 70 kDa (Hsp70) in neural and glial brain cells has constitutional and inductive options. Thus, Hsp70 expression is significantly upregulated in ischemic conditions [29]. In this study, analysis of the intensity of immunofluorescent brain cortex staining with antibodies against Hsp70 revealed a different response of brain cells in the area of ischemic damage in experimental animals when compared with intact animals (Figure 3A,G). The Hsp70 expression level in control (NaCl) (Figure 3B,G) and Ad5-GFP (Figure 3C,G) groups was 27.375 [24.940; 28.649] and 34.361 [31.315; 35.612], respectively, and was significantly higher than that of intact animals (18.422 [17.786; 19.510]). In the therapeutic groups with preventive gene therapy with Ad5-VEGF-GDNF-NCAM (20.419 [18.510; 22.950]) (Figure 3E,G) and UCB-MC+Ad5-VEGF-GDNF-NCAM (18.169 [17.462; 18.964]) (Figure 3F,G), Hsp70 immunoexpression was lower compared to the control groups (*p* < 0.05) and did not differ from intact group (*p* < 0.05). In the UCB-MC+Ad5-GFP group (Figure 3D,G), the expression level of Hsp70 (17.651 [17.648; 20.832]) was lower than the control groups and did not differ from the intact and therapeutic groups.

Activation of Caspase3 enzyme characterizes the irreversible phase of nuclear DNA degradation and subsequent cell death. In the present study, the activity of the pro-apoptotic protein Caspase3 was studied in the ischemic brain cortex in experimental rats (Figure 4A–F). It was found that the therapeutic groups Ad5-VEGF-GDNF-NCAM (12.00 [9.00; 13.00]) (Figure 4D,F) and UCB-MC+Ad5-VEGF-GDNF-NCAM (9.00 [7.50; 9.00]) (Figure 4E,F) had significantly less Caspase3-positive cells than the control NaCl (25.00 [22.00; 27.00]) group (Figure 4A,F). The number of apoptotic cells in the UCB-MC+Ad5-VEGF-GDNF-NCAM group was also lower than the Ad5-GFP control group (15.50 [11.75; 19.00]) (Figure 4B,F) (*p* < 0.05). In the UCBC+Ad5-GFP group (6.00 [3.25; 9.75]) (Figure 4C,F), the content of Caspase3-positive cells was lower than in the control groups (*p* < 0.05) and did not differ from other therapeutic groups. 

#### 2.3.2. Neuroglia Cells

Analysis of GFAP-positive cells (astrocytes), Olig2-positive cells (oligodendrocytes), and Iba1-positive cells (microglia) revealed different reactions of neuroglia cells in the ischemic area of the brain in experimental animals.

Microglial cells are the dominant cell type involved in post-stroke neuroinflammation and the organization of the glial scar. In the intact group, the immunopositive area for microglia cells detected with Iba1 marker was lower (5.42 [4.96; 5.98]) (Figure 5A,G) than in the control NaCl (12.57 [10.31; 16.82]) (Figure 5B,G) and Ad5-GFP (14.43 [12.93; 16.01]) (Figure 5C,G) groups and in the therapeutic UCB-MC+Ad5-GFP (10.07 [7.67; 10.52]]) group (Figure 5D,G) (*p* < 0.05). The Iba1-positive area in the therapeutic Ad5-VEGF-GDNF-NCAM (5.60 [4.89; 6.69]) (Figure 5E,G) and UCBC+Ad5-VEGF-GDNF-NCAM (8.80 [6.37; 9.99]) (Figure 5F,G) groups was lower when compared with the control groups and did not differ from the intact group (*p* < 0.05).

Astrocytes in the post-ischemic brain, in association with microglial cells, are involved in the formation of the glial scar, with physical and chemical properties inhibitory for brain recovery. Decreased astrogliosis often correlates with reduced volume of ischemic infarct [30]. The GFAP-positive areas in the brains of the intact group was lower (4.12 [3.36; 4.85]) (Figure 6A,G) than in the control NaCl (9.34 [8.08; 10.15]) (Figure 6B,G) and Ad5-GFP (8.92 [6.25; 12.42]]) (Figure 6C,G) groups (*p* < 0.05). In the therapeutic groups Ad5-VEGF-GDNF-NCAM (4.22 [3.36; 5.19]) (Figure 5E,G) and UCBC+Ad5-VEGF-GDNF-NCAM (3.43 [2.81; 4.19]) (Figure 5F,G), the GFAP-positive areas were lower than in the control (NaCl and Ad5-GFP) groups and did not differ from intact group (*p* < 0.05). In the UCBC+Ad5-GFP (Figure 5D,G) group, the GFAP-positive area did not differ in comparison with all experimental groups (*p* < 0.05). 

Postischemic brain damage is accompanied by the destruction of oligodendrocytes and the subsequent demyelination of neural processes [31]. The number of Olig2-positive cells was significantly reduced in the control NaCl (10.00 [9.00; 11.00]) (Figure 7B,G) and Ad5-GFP (10.00 [9.00; 11.00]) (Figure 7C,G) groups and in the UCBC+Ad5-GFP group (9.00 [9.00; 11.00]) (Figure 7D,G) relative to the intact group (14 [13; 19]) (Figure 7A,G) (*p* < 0.05). The number of Olig2-positive cells in the intact group did not differ from the therapeutic Ad5-VEGF-GDNF-NCAM (12.00 [9.00; 18.00]) (Figure 7E,G) and UCBC+Ad5-VEGF-GDNF-NCAM (14.00 [12.00; 16.00]) (Figure 7F,G) groups. It is important to note that there were more oligodendrocytes in the UCBC+Ad5-VEGF-GDNF-NCAM group than in the control groups and in the UCBC+Ad5-GFP group (*p* < 0.05), but the number did not differ from the Ad5-VEGF-GDNF-NCAM group.

#### 2.3.3. Synaptic Proteins

Synaptothysin is associated with synaptic vesicles involved in nerve impulse transmission, and its content in the presynaptic area reflects the functional activity of neurons. The synaptophysin level of fluorescence was reduced in the control NaCl (55.00 [53.00; 61.00]) (Figure 8B,G) and Ad5-GFP (56.00 [48.00; 59.00]) (Figure 8C,G) groups when compared with the intact group (78.00 [61.00; 87.00]) (Figure 8A,G) (*p* < 0.05). Immunoexpression of synaptophysin in the therapeutic Ad5-VEGF-GDNF-NCAM (64.00 [61.00; 72.00]) (Figure 8E,G), UCBC+Ad5-VEGF-GDNF-NCAM (62.00 [61.00; 65.00]) (Figure 8F,G) and UCBC+Ad5-GFP groups (79.00 [76.00; 81.00]) (Figure 8D,G) was higher than in the control groups and did not differ from the intact group (*p* < 0.05).

Postsynaptic density protein 95 kDa (PSD95) is associated with the functional activity of the postsynaptic membrane, providing its stability and plasticity in interactions with neurotransmitters. As for the intact group, the immune expression of PSD95 (65.0 [62.00; 69.00]) (Figure 9A,G) was reduced in all experimental groups except for the group UCBC+Ad5-VEGF-GDNF-NCAM (55.00 [50.00; 57.00]) (*p* < 0.05) (Figure 9F,G). However, the level of PSD95 fluorescence was significantly higher in the therapeutic Ad5-VEGF-GDNF-NCAM (62.50 [57.75; 67.00]) (Figure 9E,G), UCBC+Ad5-VEGF-GDNF-NCAM and UCBC+Ad5-GFP (50.00 [48.50; 57.50]) (Figure 9D,G) groups, in comparison with the control group NaCl (43.00 [38.50; 46.00]) (Figure 9B,G), which did not differ from the Ad5-GFP group (49.00 [46.00; 51.00]) (Figure 9C,G) (*p* < 0.05).

### 2.4. Expression of Recombinant Molecules in the Rat Brain after Intrathecal Adenoviral-Mediated Delivery of Transgenes

The expression of the reporter *gfp* and therapeutic *vegf165*, *gdnf* and *ncam1* genes was studied in the stroke area of rat brains, 25 days after an intrathecal injection of Ad5-GFP or Ad5-VEGF-GDNF-NCAM. Green fluorescent protein was detected in brain cells by laser microscopy (Figure 10A). Immunofluorescent analysis using specific antibodies to VEGF, GDNF, and NCAM also revealed target molecules in the brain cells (Figure 10B–D). In the peri-infarct zone, we observed an approximately equal distribution of the cells producing reporter GFP or recombinant VEGF, GDNF, and NCAM molecules.

### 2.5. Expression of Recombinant Molecules in Rat Brain after Intrathecal UCB-MC-Mediated Delivery of Transgenes

The synthesis of the reporter GFP in UCB-MC in vivo was studied on day 24 after intrathecal injection of UCB-MC transduced with Ad-GFP. Antibody to human nuclear antigen (HNA) was used for the identification of UCB-MC in the rat brain cortex. HNA-positive cells producing GFP were revealed in the peri-infarct zone (Figure 11A). The production of therapeutic molecules (VEGF, GDNF, and NCAM) in transplanted, genetically-modified UCB-MC was studied using a double immunofluorescent staining method with antibodies against HNA and to one of the recombinant molecules. HNA-positive cells producing recombinant human molecules VEGF (Figure 11B), GDNF (Figure 11C), and NCAM (Figure 11D) were detected in the left hemisphere, 24 days after the intrathecal injection of UCB-MC+Ad5-VEGF-GDNF-NCAM.

### 2.6. Multiplex Analysis of Cytokines, Chemokines and Growth Factors in Blood (Serum) and Cerebrospinal Fluid (Liquor) Samples of Experimental and Intact Animals

Granulocyte colony-stimulating factor (G-CSF), granulocyte and monocyte colony-stimulating factor (GM-CSF), interferon-γ (IFN-γ), interleukin-4 (IL-4), interleukin-10 (IL-10), interleukin-12 (IL-12), and VEGF are considered cytokines with neuroprotective action, while interleukin-1 (IL-1), interleukin-6 (IL-6), interleukin-17 (IL-17), and tumor necrosis factor-α (TNF-α) can cause secondary death of nerve cells. In the present study, preliminary analysis of endogenous cytokines, chemokines and growth factors in cerebrospinal fluid in animals from the control group NaCl showed no significant differences in comparison with intact rats, which probably indicates the cessation of inflammatory response 21 days after modelling a stroke. In addition, intrathecal injection of a combination of adenoviral vectors carrying cDNA encoding VEGF, GDNF, and NCAM, or genetically-modified UCB-MC+Ad5-VEGF-GDNF-NCAM on day 21 of the experiment does not reliably affect the cytokine levels in liquor samples when compared with intact rats.

## 3. Discussion

It is known that ischemic stroke leads to massive loss of neurons and glial cells immediately after acute arterial occlusion. Neuron death caused by necrosis and apoptosis is the result of blood flow impairment, oxidative stress, mitochondrial dysfunction, and excitotoxicity. Unfortunately, these irreversible changes rapidly spread beyond the epicenter of the ischemic damage, resulting in progressive neurodegeneration on the following days, and even over the following week [32]. However, during the first hours, an ischemic penumbra with blood supply below a normal level is formed around the ischemic core with irreversible changes. The viability of cells in the penumbra is limited to a 3–6-h period of the therapeutic window, during which it is possible to restrain the entry into apoptosis of neurons and glial cells and thus prevent the increase of infarction volume.

Despite the improved efficiency of modern therapy for ischemic brain stroke, recovery of patients is not satisfactory. Today, specific therapy for acute stroke is aimed at restoring blood circulation in the ischemic area and maintaining the metabolism of brain cells. However, the drugs used in practical medicine are not effective in controlling the death of nerve cells in the ischemic penumbra, which results in an extensive increase in brain infarction. It is known that there are natural limitations neuroregeneration in the CNS; therefore, there is a need to develop new effective approaches to maintain the viability of nerve cells in the ischemic penumbra in the 3–6-h therapeutic window.

It is known that neuroontogenesis (addressed migration of brain cells, directed axon growth and establishment of intercellular contacts) involves an information exchange between neurons, which is realized through neurotrophic factors produced in some neurons or in non-neural cells that affect other neurons. In postnatal ontogenesis, there is no formation of new cortical neurons. Therefore, dying neurons are not restored. This does not mean that there is no regeneration in brain cortex; however, as regeneration is carried out due to intracellular regeneration of damaged neurons, growth of their neuritis and restoration of synaptic connections between regenerating and intact neurons. During neuroregeneration, neurotrophic factors support neuronal survival, stimulate axonal growth, and establish lost synaptic contacts.

At present, the most promising of the actively developed strategies to prevent brain cell death in penumbra is the development of gene and cellular technologies. Gene therapy is mainly aimed at the delivery of therapeutic genes encoding neurotrophic factors. Among them, the most promising are genes encoding neurotrophic factors (BDNF, CNTF, GDNF, VEGF), anti-apoptotic proteins (Bcl-2, Bcl-XL), heat shock proteins (Hsp25, Hsp70), and anti-inflammatory molecules (IL-1RA). The neuroprotective effect of these factors has been experimentally proven, but there is no unequivocal, let alone exhaustive, answer to the question of which of these factors may be recommended as neuroprotective factors in practical medicine. It has been established that combinations of several neurotrophic factors may have a more pronounced effect on nerve cell survival [25,28].

Other equally important issues in the strategy for the gene therapy of stroke include the development of technology to deliver transgenes to the brain. Difficulties in the delivery of therapeutic genes to the cerebral infarction area are one of the main reasons for the unavailability of effective gene therapy in treating post-ischemic negative consequences in the brain. Direct (in vivo) gene therapy provides for the delivery of transgenes into the recipient brain plasmid or viral genetic vectors [33]. Cell-mediated (ex vivo) gene therapy is based on the delivery of therapeutic genes using cells that serve as carriers of transgenes, as well as producers of recombinant protein molecules.

For the treatment of stroke, various methods of gene therapy are known, the effectiveness of which has been proven in numerous experiments on animals [2,4,34]. Injections of viral vectors carrying transgenes into the ventricles or the infarction area are mainly used to deliver the therapeutic genes to the brain. Genetic vectors based on Sendai virus vectors containing *gdnf* or *ngf* genes were injected 30 min after stroke simulation [35]. The adeno-associated viral vector carrying *gdnf* was injected 2 days after the stroke modelling [36]. In other studies, viral vectors carrying therapeutic genes were delivered before modelling the stroke. The positive effect was shown after local delivery to the brain of *gdnf* or *cntf* genes—7 days [37], *bdnf*—2 weeks [38], *gdnf*—4 weeks [39], or *ngf* and *bdnf*—4 to 5 weeks [40] before stroke modelling.

The list of genes employed in gene therapy for stroke is quite long. Of these, we find *vegf*, *gdnf*, and *ncam* to be the most promising.

In addition to angiogenic action, VEGF exhibits the properties of a typical neurotrophic factor. It supports survival of sensitive [41] and motor neurons [42] and stimulates proliferation of astrocytes [43], neural stem cells [44], and Schwann’s cells [41]. GDNF has a pronounced neuroprotective effect on dopaminergic brain neurons and cholinergic spinal cord motoneurons [45] and stimulates the growth of nerve processes [46]. NCAM (CD56) is expressed on the surface of neurons and glial cells. Intercellular interactions mediated by NCAM in neuro-ontogenesis and posttraumatic regeneration provide survival and migration of neurons, directed neurite growth and synaptogenesis.

The efficiency of cell therapy for stroke treatment in experiments using neural precursors derived from embryonic stem cells [47], induced pluripotent cells [48], MSCs isolated from red bone marrow [49] or UCB [50] suggests the use of these cells as carriers of therapeutic genes for delivery to the brain. Thus, MSCs were used for delivery of *bdnf* [51], *pigf* [52], and *vegf165* [53] to the brain. Of particular interest is the transduction of cell carriers by two or more expression vectors [28]. This approach allows simultaneous overexpression of several molecules/stimulants of neuroregeneration to be obtained. In our studies, in addition to the gene encoding neurotrophic factors, a gene encoding NCAM was delivered into UCB-MC, which, according to the obtained data, promoted the addressed migration of transplanted cells into the CNS after intravenous injection, increased their survival in the recipient’s tissues and supported prolonged production of recombinant therapeutic molecules [28]. In clinical investigations, autologous cells isolated from the red bone marrow (mononuclear and MSCs) or peripheral blood (CD34+) are predominantly used in the cell therapy of patients after stroke [54,55].

The most promising cell carriers of therapeutic genes are UCB-MC [56,57,58]. The basis for their application is the suitability for both allografting and autotransplantation in humans, availability, and the ease of obtaining and storage. An important factor is the absence of legal, ethical, and religious prohibitions related to blood cell transplantation. In our previous study, we showed that intrathecal injection of adenoviral vectors carrying *vegf*, *gdnf*, and *ncam*, or genetically-modified UCB-MC+Ad5-VEGF-GDNF-NCAM, 4 h after stroke modelling in rats, had a positive effect on the morpho-functional recovery of the post-ischemic brain [25]. Adenoviral vectors and genetically-modified UCB-MC with cerebrospinal fluid reached the ischemic area and delivered the production of recombinant VEGF, GDNF, and NCAM, lasting up to 21 days in the experiment. 

Other important issues in the strategy for the treatment of ischemic stroke include the development of approaches to enhance the viability of neurons with the threat of a stroke. Patients with transient ischemic attacks, arterial hypertension, atrial fibrillation, disorders of lipid metabolism with high cholesterol and diabetes are at high risk of ischemic stroke. Preventive therapy aimed at increasing the survivability of neurons in at-risk patients may prevent severe post-ischemic consequences in the brain, or improve the outcome of the disease. Currently, in medical practice, measures to prevent stroke are based on the use of anticoagulants and prosthetics of blood vessels. At the same time, the preventive methods that are able to considerably decrease the death of neurons in the “ischemic penumbra” during the 3–6-h “therapeutic window” are unknown. Enhancement of the viability of nerve cells at risk of stroke is also associated with the delivery of therapeutic genes that encode molecules to the brain, which inhibit neuronal death and stimulate neuroregeneration. In this study, for the first time, we propose the approach of preventive gene therapy to improve the viability of brain neurons under threat of ischemic stroke to contain neuronal death in the first hours of a stroke. The use of leucocytes for delivery of therapeutic genes (*vegf165*, *gdnf*, and *ncam1*) in the brain was based on their biological properties. Leucocytes are cells with high secretory and migration potentials, which suggest their exclusive role as cell carriers for addressed delivery and effective expression of transgenes. The results obtained in the study demonstrate that preventive intrathecal adenoviral- or UCB-MC-mediated delivery of *vegf165*, *gdnf*, and *ncam1* results in a reduction of apoptosis and, consequently, the infraction volume. In addition to the decrease in expression of proteins of cellular stress and restraining neuronal death in the area of ischemic damage, we found evidence of the restoration of functional activity of neurons (increase in expression of synaptic proteins), maintenance of myelinization (increase in the number of oligodendrocytes) and an obstacle to astrogliosis development (decrease in the immunopositive areas for astrocytes and microglial cells markers). Importantly, transplantation of gene-modified UCB-MC is safer and more efficacious compared with direct gene therapy. These data are in line with our results using the same gene and gene-cell constructs for ischemic stroke treatment in rats and allow us to conclude that preventive gene therapy may be effective in overcoming the negative consequences of ischemic stroke in the rehabilitation period.

Recently, for personalized ex vivo gene therapy, we suggested the use of gene-modified leucoconcentrate (GML) prepared from patient’s peripheral blood and chimeric adenoviral vectors (Ad5/35F) carrying one or a combination of therapeutic genes [59]. Taken together with the concept of GML-therapy and the data of this study, we propose the use of GML carrying *vegf165*, *gdnf*, and *ncam1* for personalized preventive gene therapy in the threat of stroke.

## 4. Materials and Methods

### 4.1. Preparation and Molecular Analysis of Gene-Modified UCB-MC

Human (*vegf165*, *gdnf*, *ncam1*) and reporter (*gfp*) genes were inserted into recombinant replication-defective adenovirus serotype 5 (Ad5) in the Gamaleya Research Institute of Epidemiology and Microbiology (Moscow, Russia), as described previously [25]. Viral vectors were grown in HEK-293 cell culture, purified by exclusion chromatography and the titres of Ad5-VEGF (2.6 × 10^9^ PFU/mL), Ad5-GDNF (1.7 × 10^10^ PFU/mL), Ad5-NCAM (2.4 × 10^10^ PFU/mL) and Ad5-GFP (1.2 × 10^10^ PFU/mL) were determined in the HEK-293 cell culture using the plaque formation technique.

The fraction of mononuclear cells from UCB was isolated onto a density barrier by the standard technique of sedimentation, seeded in 10 cm culture dishes and simultaneously transduced with three adenoviral vectors at an equal ratio: Ad5-VEGF (1/3)+Ad5-GDNF (1/3)+Ad5-NCAM (1/3) or with Ad5-GFP with MOI of 10, as described previously [25]. To confirm the efficacy of UCB-MC transduction, the gene-modified cells were cultured for 96 h. Transduction with therapeutic genes (*vegf165*, *gdnf*, *ncam1*) was confirmed by a real-time polymerase chain reaction (RT-PCR) and with reporter green fluorescent protein (*gfp*) using fluorescent microscopy, as described previously [57].

For direct (in vivo) preventive gene therapy, a mixture of 2 × 10^7^ virus particles in an equal ratio of Ad5-VEGF (1/3)+Ad5-GDNF (1/3)+Ad5-NCAM (1/3) in 20 μL of saline and Ad5-GFP in 20 μL of saline were prepared. For cell-mediated (ex vivo) preventive gene therapy, 2 × 10^6^ UCB-MC+Ad5-VEGF+Ad5-GDNF+Ad5-NCAM in 20 µL of saline and 2 × 10^6^ UCB-MC+Ad5-GFP in 20 µL of saline were prepared.

The efficiency of UCB-MC genetic modification by adenoviral vector, bearing green fluorescent protein reporter gene, was analyzed 72 h after cell transduction. The production of green fluorescent protein UCB-MC+Ad5-GFP was examined with the Axio Oberver Z1 inverted fluorescence microscope (Carl Zeiss, Germany). The number of transduced UCBC was estimated using a BD FACSAria III flow cytomography fluorimeter (BD Bioscience, New York, NY, USA) and BD FACS Diva7 software (BD Bioscience, New York, NY, USA). 

mRNA levels of *vegf165*, *gdnf*, and *ncam1* transgenes in the UCB-MC, simultaneously transduced by a combination of three adenoviral vectors Ad5-VEGF, Ad5-GDNF and Ad5-NCAM, were analyzed with PCR-RT. Extraction of common RNA from UCB-MC+Ad5-VEGF-GDNF-NCAM and UCB-MC+Ad5-GFP, 72 h after cell incubation, was carried out using a set Yellow Solve (Silex), according to the manufacturer’s instructions. Reference samples of common RNA were derived from non-transduced UCBC. The cDNA synthesis was performed using six nucleotide random primers and reverse transcriptase RevertAid Reverse Transcriptase (Thermo Fisher Scientific). Quantitative level analysis mRNA of the target cDNA was performed using a CFX 96 Real-Time PCR System thermocirculator (BioRad, Hercules, CA, USA), using the TaqMan system. Sequences of primers and probes of the reaction mixture are presented in Table 1. Received data was normalized by the 18S rRNA reference gene. For each target gene, the results were obtained in two independent experiments and presented as a mean value ± SE, *p* < 0.05.

Supernatant obtained 72 h after incubation of gene-modified UCB-MC (UCB-MC+Ad5-VEGF-GDNF-NCAM and UCB-MC+Ad5-GFP) and intact UCB-MC were used for multiplex profiling. In this work, commercially available panels, Human Cytokine/Chemokine Magnetic Bead Panel (HCYTMAG-60K-PX41), were applied, containing the following analytes: EGF, FGF2, Eotoxin, TGF-a, G-CSF, Flt-3L, GM-CSF, Fractalkine, IFNa2, IFN-g, GRO, IL10, MCP-3, IL12-p40, MDC, IL12-p70, PDGF-AA, IL-13, PDGF-AB/BB, IL-15, sCD40L, IL-17a, IL-1ra, IL-1a, IL-9, IL-1bIL-2, IL-3, IL-4, IL-5, IL-6, IL-7, IL-8, IP10, MCP-1, MIP-1a, MIP-1b, RANTES, TNFa, TNFb, and VEGF (Millipore). The quantitative analysis of target molecules was performed on a Bio-Plex200 System (BioRad, USA) according to the protocol recommended by the manufacturer and using Bio-Plex Manager 4.1 software (Bio-RadLaboratories). The results were statistically processed using the following methods, using linear models implemented in the limma package. Fold Change was used as an estimate for the effect size, obtained for pairwise comparison.

### 4.2. Animals and Treatments

The animal protocols were conducted in strict compliance with the guidelines established by the Kazan State Medical University Animal Care and Use Committee (approval No. 10, 2017). Experiments were performed on adult female Wistar rats (weight of 200–250 g) obtained from Pushchino Laboratory (Pushchino, Russia). Animals were housed one per cage according to approved procedures for the use of animals in laboratory experiments.

For preventive gene therapy, rats were deeply anesthetized intraperitoneally with Zolitil 100 (Virbac Laboratoires, France) 3 mg/kg and Xyla (Interchemie werken “De Adelaar” B.V., Netherlands) 4.8 mg/kg. A laminectomy was made over the L4–L5 vertebral level and gene-modified UCBC or adenoviral vectors were infused intrathecally. The rats were divided into five experimental groups according to the injected substances (Table 1).

Ischemic stroke in the rats was induced 3 days after intrathecal injection of UCB-MC+Ad5-GFP and UCB-MC+Ad5-VEGF-GDNF-NCAM or 4 days after intrathecal injection of 0.9% NaCl, Ad5-GFP and Ad5-VEGF-GDNF-NCAM. The ischemic stroke was induced by permanent MCAO, as described previously [25]. In brief, to reduce blood flow in the circle of Willis, the right common carotid artery was ligated with surgical silk. In the left side, a 4–5 mm hole was drilled in the temporal bone and MCAO was performed by thermocoagulation using an operating microscope. 

To reduce suffering and distress, post-operative care included analgesic therapy with Ketamine (Dr. Reddy’s Laboratories, Ltd., Hyderabad, Andhra Pradesh, India) intramuscularly (2.5 mg/kg) once-a-day and antibacterial therapy with Ceftriaxone (Sandoz, Austria) intramuscularly (50 mg/kg) once-a-day for 5 days. Based on our previous study [25] for evaluation of preventive gene therapy, experimental animals were sacrificed three weeks after MCAO.

### 4.3. Morphometric and Immunofluorescent Analysis of the Brain

For histological analysis, rats were deeply anesthetized by intraperitoneal injection of sodium pentobarbital (60 mg/kg) and intracardially perfused with 4% paraformaldehyde (PFA, Sigma) in phosphate-buffered saline (PBS, pH 7.4). The brains of the rats were isolated from the skull, post-fixed in 4% paraformaldehyde and cryoprotected in 30% sucrose. Frozen frontal sections of the brain were obtained through the epicenter of ischemic injury using a cryostat (Microm HM 560, Thermo Scientific, Waltham, MA, USA).

Morphometric analysis of the volume of the cerebral infarction included macroscopic evaluation of the infarct area of the whole brain by capturing digital images and microscopic study every 10 frontal brain sections (with 200 µm interval) through the ischemic injury area, after staining with hematoxylin and eosin. To calculate the infarction volume, maximal depth and maximal diameter of the infarct cavities were estimated using ImageJ software (NIH). The calculation was carried out as V = 1/3 × maximal depth × π × (maximal diameter/2)**^2^**.

Immunofluorescent staining was performed on 20 μm sections of frontal brain using primary antibodies (Ab) reacting with cell specific markers (Table 2). The viability of neural cells was evaluated using Ab to pro-apoptotic protein (Caspase3) and heat shock protein 70 kDa (Hsp70). Synaptic function of preserved neurons was assessed with Ab against synaptophysin and postsynaptic density protein 95 kDa (PSD95). Responses of glial cells were assessed with Ab to GFAP for astrocytes, Ab to Olig2 for oligodendrocytes and Ab to Iba1 for microglia. Anti-human nuclear antigen (HNA) antibodies were used for identification of human UCBC in the rat brain. Double immunofluorescent staining with Ab to HNA and Ab to human VEGF, GDNF, and NCAM were employed to analyze the expression of those recombinant molecules in UCB-MC. For immunofluorescent labelling, sections were incubated with secondary antibodies (Table 2). Appropriate secondary Alexa Fluor 647 and 488 conjugated Abs were used for immunofluorescent labelling of the target molecules. Cell nuclear counterstaining was performed with DAPI (10 μg/mL in PBS, Sigma), sections were embedded in glycerol (GalenoPharm, Saint Petersburg, Russia) and observed under a LEICA TCS SP5 MP microscope (Leica Microsystems, Wetzlar, Germany).

Digital images were captured using identical confocal settings in areas of 0.05 mm^2^ and analyzed with ImageJ software (NIH) in 10 captured fields in the peri-infarct zone (Figure 2C) in each section, as described previously [25]. The number of immunopositive cells for apoptosis protein Caspase3 and oligodendroglial cells transcription factor Olig2 were counted in regard to nuclear counterstaining with DAPI. Neuroglial markers for astrocytes (GFAP) and microglia (Iba1) were evaluated as the immunopositive areas and presented in percentages. The level of synaptic proteins (synaptophysin and PSD95) and Hsp70 immunoexpression in brain sections was evaluated as mean pixel intensities.

### 4.4. Multiplex Cytokine Analysis of the Cerebrospinal Fluid

Multiplex profiling was performed in cerebrospinal fluid (liquor) of experimental and intact animals using the commercially available Bio-Plex Pro™ Rat Cytokine 23-Plex Assay panel, containing the following analytes: G-SCF, GM-CSF, GRO/KC, IFN-g, IL-10, IL-12p70, IL-13, IL-18, IL-1a, IL-1b, IL-2, IL-4, IL-5, IL-6, IL-7, IL17A, M-CSF, MCP-1, MIP1a, MIP3a, RANTES, TNFa, and VEGF. Quantitative analysis of target molecules was performed on a Bio-Plex200 System analyzer (BioRad, USA), as recommended by the manufacturer, using Bio-Plex Manager 4.1 software (Bio-RadLaboratories). The statistical processing of obtained results was carried out using linear models implemented in the limma package.

### 4.5. Statistics

Statistical data analysis and visualizations were performed using R version 3.5.3 (R Foundation for Statistical Computing, Vienna, Austria). Sample distributions of quantitative values were visualized using box plots, and descriptive statistics are presented as: (Median [1st quartile; 3rd quartile]). The Kruskal–Wallis test was used to compare morphometric and immunofluorescent staining data between experimental groups when a null hypothesis was rejected; we used Dunn’s-test as the post hoc method. Differences were considered statistically significant where *p* < 0.05.

## 5. Conclusions

Methods of preventive gene therapy of ischemic stroke in at-risk patients are currently unknown. In this study, for the first time, we provide evidence of the beneficial effects of preventive triple gene therapy by an adenoviral- or UCB-MC-mediated intrathecal simultaneous delivery combination of *vegf165*, *gdnf*, and *ncam1* on the preservation and recovery of the brain in rats with subsequent modelling of stroke. The obtained data represent a novel and potentially successful approach for ischemic stroke treatment in patients.

## Figures and Tables

**Figure 1 ijms-21-06858-f001:**
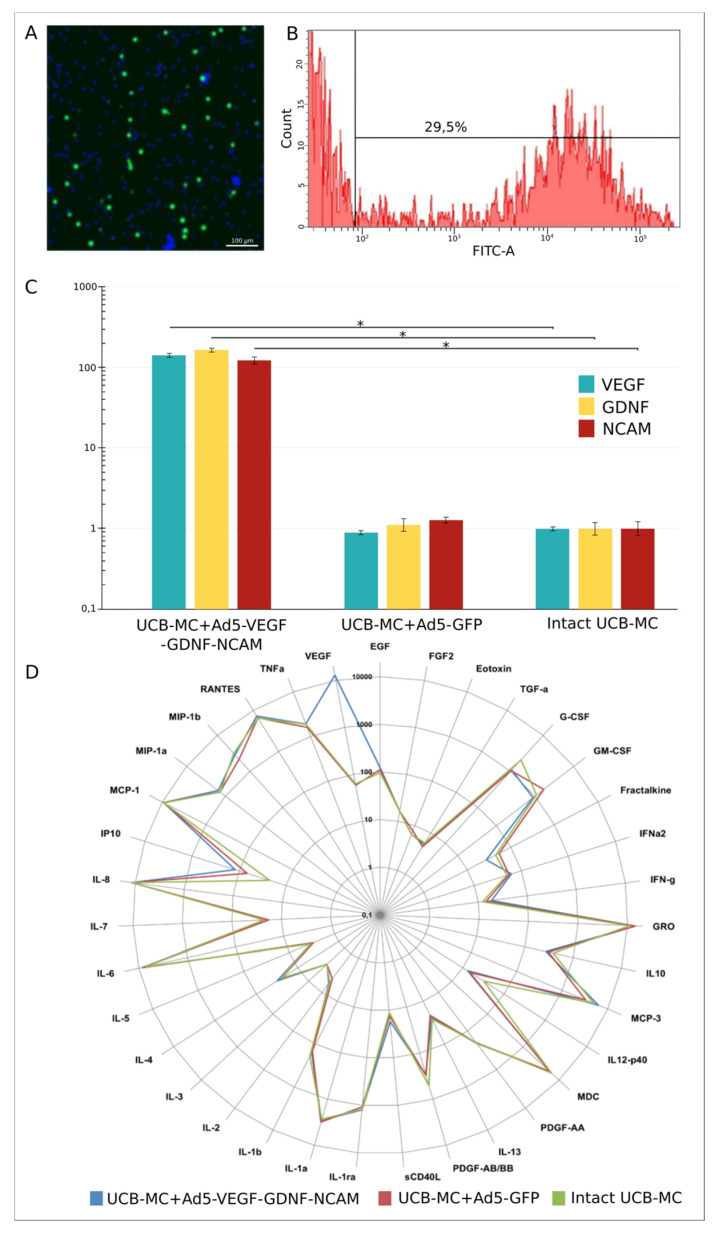
Molecular analysis of gene modified umbilical cord blood mononuclear cells (UCB-MC). (**A**,**B**)—Production of green fluorescent protein (GFP) in UCB-MC, 72 h after transduction with Ad5-GFP (MOI = 10). (**A**)—fluorescent microscopy shows a specific green glow in UCB-MC+Ad5-GFP. Cell nuclei are stained with Hoechst 33342 (blue glow). Scale bar = 200 µm. (**B**)—flow cytometry analysis demonstrates that 29.5% of UCB-MC effectively produce reporter GFP. (**C**)—Quantitative analysis of mRNA *vegf165*, *gdnf*, and *ncam1* levels in intact (naïve) UCB-MC and genetically-modified UCB-MC, 72 h after transduction with three adenoviral vectors simultaneously carrying the therapeutic genes (Ad5-VEGF, Ad5-GDNF, and Ad5-NCAM) or with Ad5-GFP (UCBC + Ad5-GFP), with a MOI = 10. Data from two independent experiments are represented as an average value of ± SE, *—*p* < 0.05. (**D**)—Radial comparative diagram of cytokine, chemokines and growth factors in supernatant obtained 72 h after incubation of gene modified UCB-MC (UCB-MC+Ad5- vascular endothelial growth factor (VEGF)- glial cell-derived neurotrophic factor (GDNF)- neural cell adhesion molecule (NCAM) and UCB-MC+Ad5-GFP) and intact UCB-MC.

**Figure 2 ijms-21-06858-f002:**
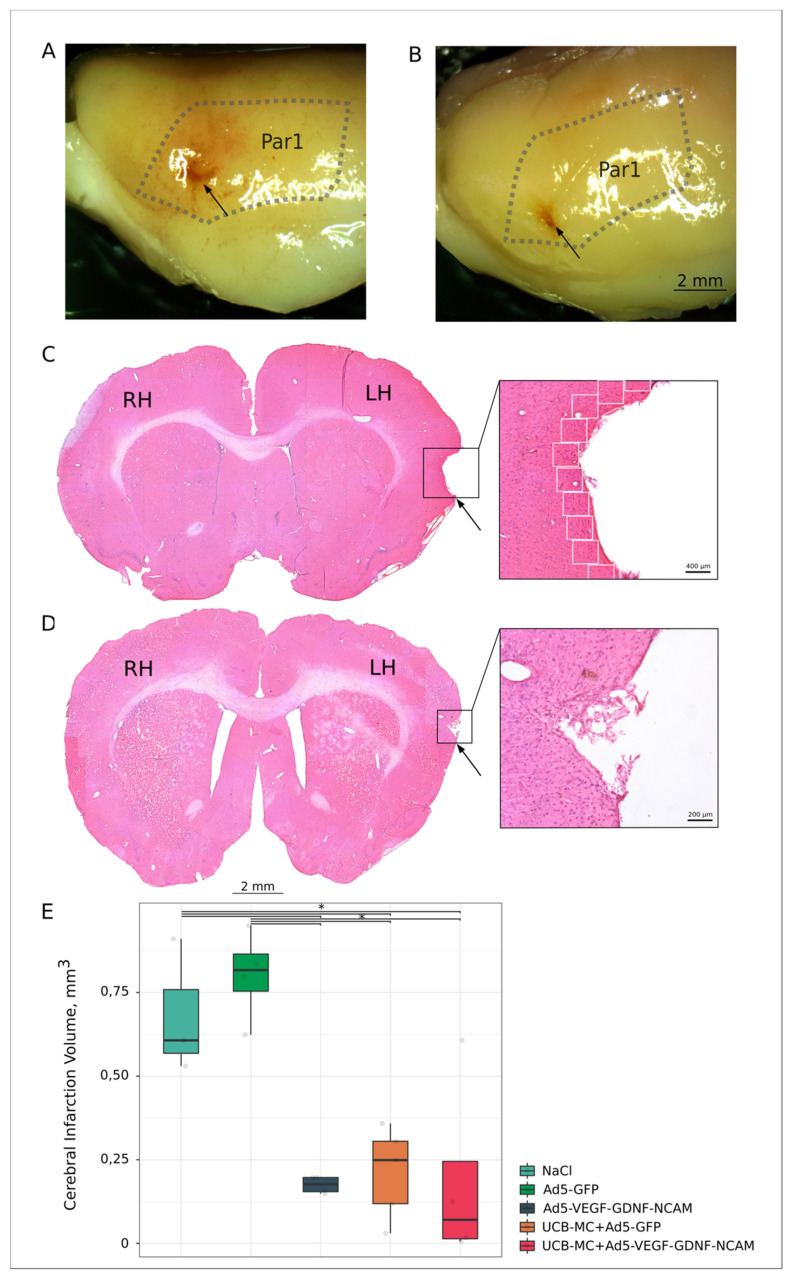
Ischemic stroke in rats 3 weeks after the distal middle cerebral artery occlusion. (**A**,**C**)—The brain from the control (NaCl) group. (**B**,**D**)—The brain from the therapeutic (UCB-MC+Ad5-VEGF-GDNF-NCAM) group. In (**A**) and (**B**) the stroke area in the parietal lobe of the brain is shown with the arrow. Par1—cortex parietals, area 1 is marked by a dotted line. In (**C**) and (**D**)—frontal sections of the brain in the stroke area stained with hematoxylin and eosin. RH—right hemisphere; LH—left hemisphere. Inserts with arrows show the stroke area with the maximum depth and the maximum radius of the infarct cavities at higher magnification. White boxes in the insert in (**C**) panel indicate areas with S = 0.05 mm^2^ in the peri-infarct zone used for immunofluorescent analysis. (**E**)—comparative morphometric analysis of the infarct cavities volume in experimental groups; *—*p* < 0.05.

**Figure 3 ijms-21-06858-f003:**
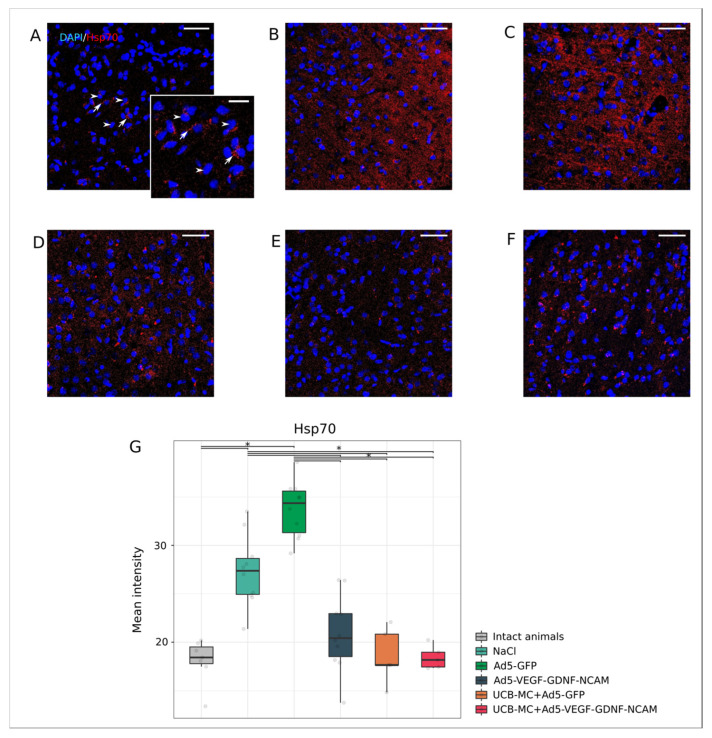
Immunoexpression of Hsp70 in the rat brain cortex, 21 days after the middle cerebral artery occlusion. Immunofluorescent staining with antibody to heat shock protein 70 kDa in intact (**A**), control NaCl (**B**) and Ad5-GFP (**C**), and therapeutic UCB-MC+Ad5-GFP (**D**), Ad5-VEGF-GDNF-NCAM (**E**) and UCB-MC+Ad5-VEGF-GDNF-NCAM (**F**) groups. Arrows indicate cytoplasmic localization of Hsp70 (red glow) in brain cells. Arrowheads point to cell nuclei visualized using DAPI (blue glow). Scale bar in (**A**–**F**) = 50 µm; scale bar in the insert = 20 µm. (**G**)—comparative analysis of fluorescence intensity level of Hsp70 in experimental groups; *—*p* < 0.05.

**Figure 4 ijms-21-06858-f004:**
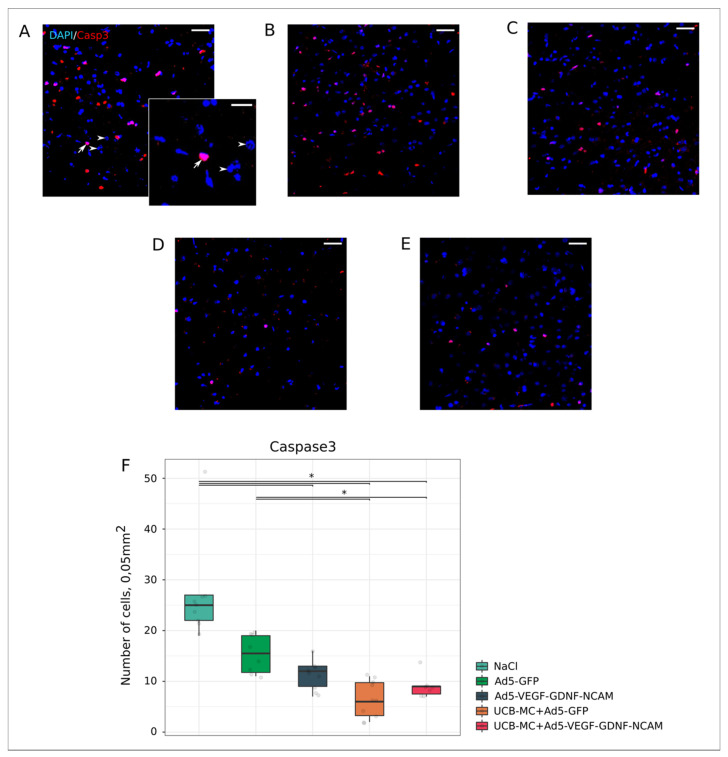
Count of Caspase3-positive cells in the rat brain cortex, 21 days after the middle cerebral artery occlusion. Immunofluorescent staining with antibody to Caspase3 in control NaCl (**A**) and Ad5-GFP (**B**), and therapeutic UCB-MC+Ad5-GFP (**C**), Ad5-VEGF-GDNF-NCAM (**D**) and UCB-MC+Ad5-VEGF-GDNF-NCAM (**E**) groups. Arrow indicates nuclear localization of Caspase3 (red glow) in brain cells. Arrowheads point to cell nuclei visualized using DAPI (blue glow). Scale bar in (**A**–**E**) = 50 µm; scale bar in the insert = 20 µm. (**F**)—comparative analysis of Caspase3-positive cells number in experimental groups; *—*p* < 0.05.

**Figure 5 ijms-21-06858-f005:**
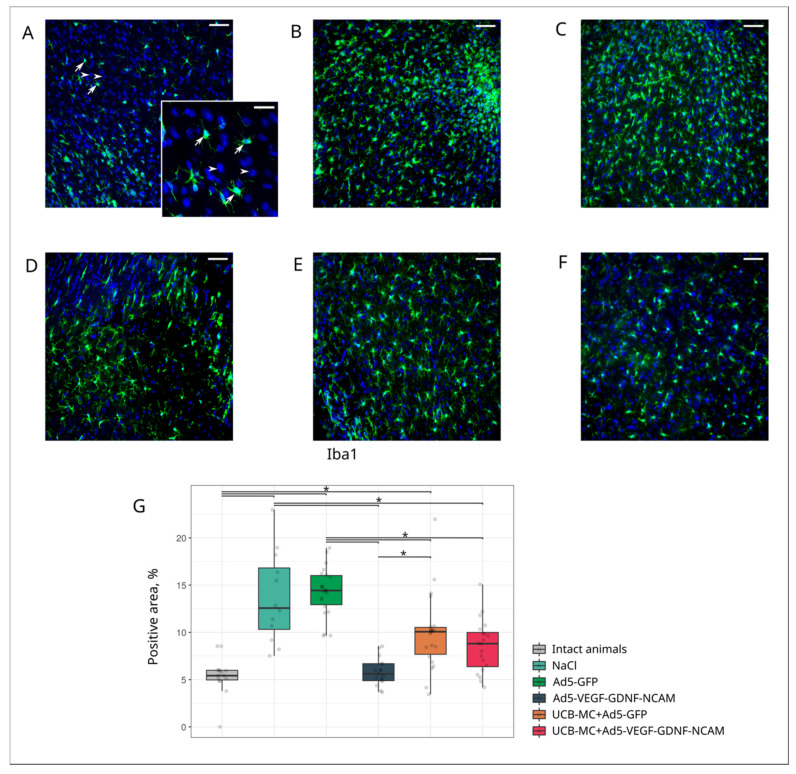
Immunoexpression of Iba1 in the rat brain cortex, 21 days after the middle cerebral artery occlusion. Immunofluorescent staining with antibody to microglia specific calcium-binding protein Iba1 in intact (**A**), control NaCl (**B**) and Ad5-GFP (**C**), and therapeutic UCB-MC+Ad5-GFP (**D**), Ad5-VEGF-GDNF-NCAM (**E**), and UCB-MC+Ad5-VEGF-GDNF-NCAM (**F**) groups. Arrows indicate cytoplasmic localization of Iba1 (green glow) in microglial cells. Arrowheads point to cell nuclei visualized using DAPI (blue glow). Scale bar in (**A**–**F**) = 50 µm; scale bar in the insert = 20 µm. (**G**)—comparative analysis of Iba1-positive areas in experimental groups; *—*p* < 0.05.

**Figure 6 ijms-21-06858-f006:**
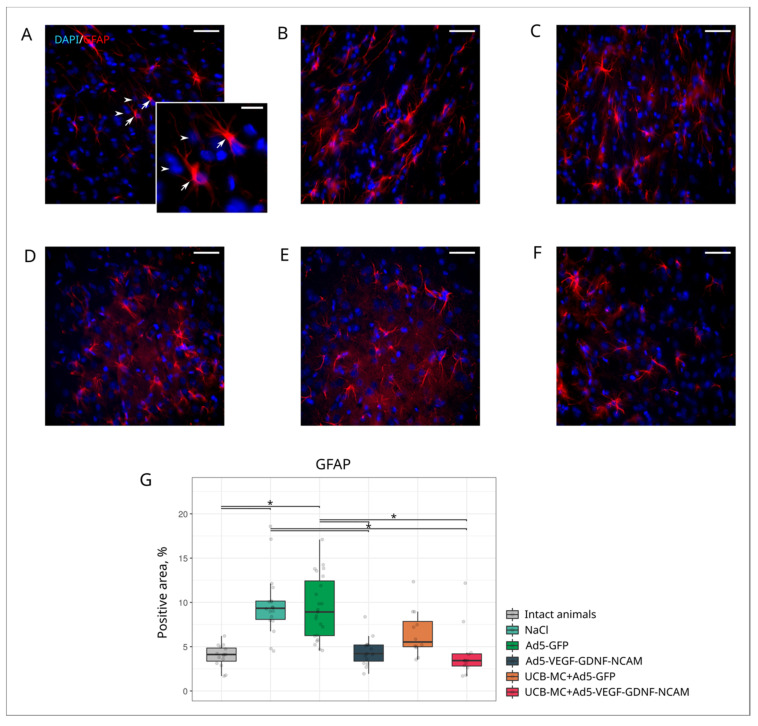
Immunoexpression of GFAP in the rat brain cortex, 21 days after the middle cerebral artery occlusion. Immunofluorescent staining with antibody to astrocyte specific cytoskeletal glial fibrillary acidic protein GFAP in intact (**A**), control NaCl (**B**) and Ad5-GFP (**C**), and therapeutic UCB-MC+Ad5-GFP (**D**), Ad5-VEGF-GDNF-NCAM (**E**) and UCB-MC+Ad5-VEGF-GDNF-NCAM (**F**) groups. Arrows indicate cytoplasmic localization of GFAP (red glow) in astrocytes. Arrowheads point to cell nuclei visualized using DAPI (blue glow). Scale bar in (**A**–**F**) = 50 µm; scale bar in the insert = 20 µm. (**G**)—comparative analysis of GFAP-positive areas in experimental groups; *—*p* < 0.05.

**Figure 7 ijms-21-06858-f007:**
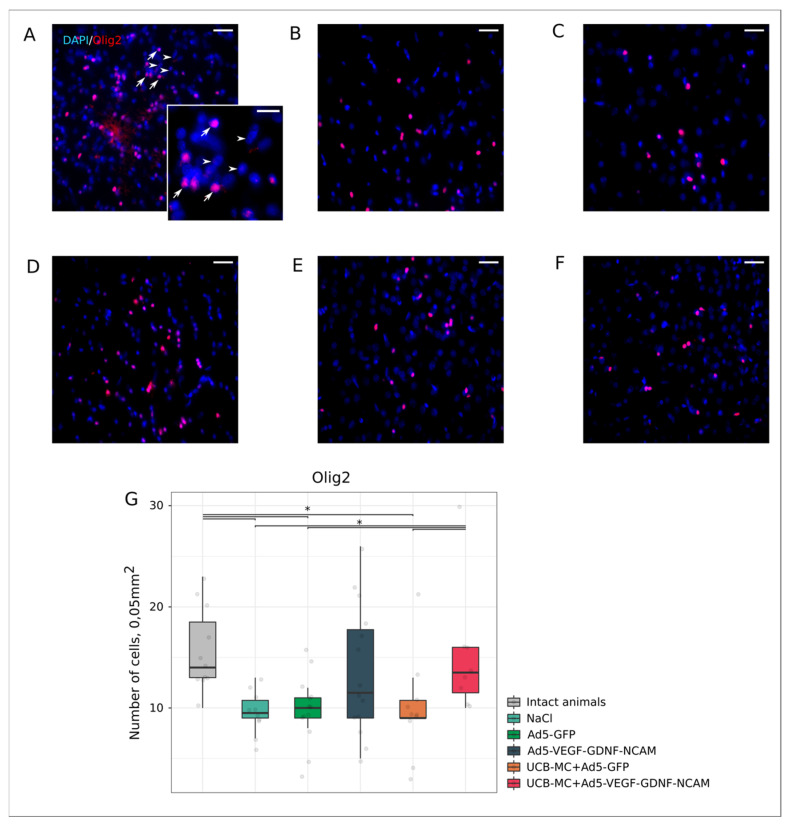
Count of Olig2-positive oligodendrocytes in the rat brain cortex, 21 days after the middle cerebral artery occlusion. Immunofluorescent staining with antibody to oligodendrocyte transcription factor Olig2 (red glow) in intact (**A**), control NaCl (**B**), and Ad5-GFP (**C**), and therapeutic UCB-MC+Ad5-GFP (**D**), Ad5-VEGF-GDNF-NCAM (**E**), and UCB-MC+Ad5-VEGF-GDNF-NCAM (**F**) groups. Arrows indicate nuclear localization of Olig2 (red glow) in oligodendrocytes. Arrowheads point to cell nuclei visualized using DAPI (blue glow). Scale bar in (**A**–**F**) = 50 µm; scale bar in the insert = 20 µm. (**G**)—comparative analysis of Olig2-positive cells number in experimental groups; *—*p* < 0.05.

**Figure 8 ijms-21-06858-f008:**
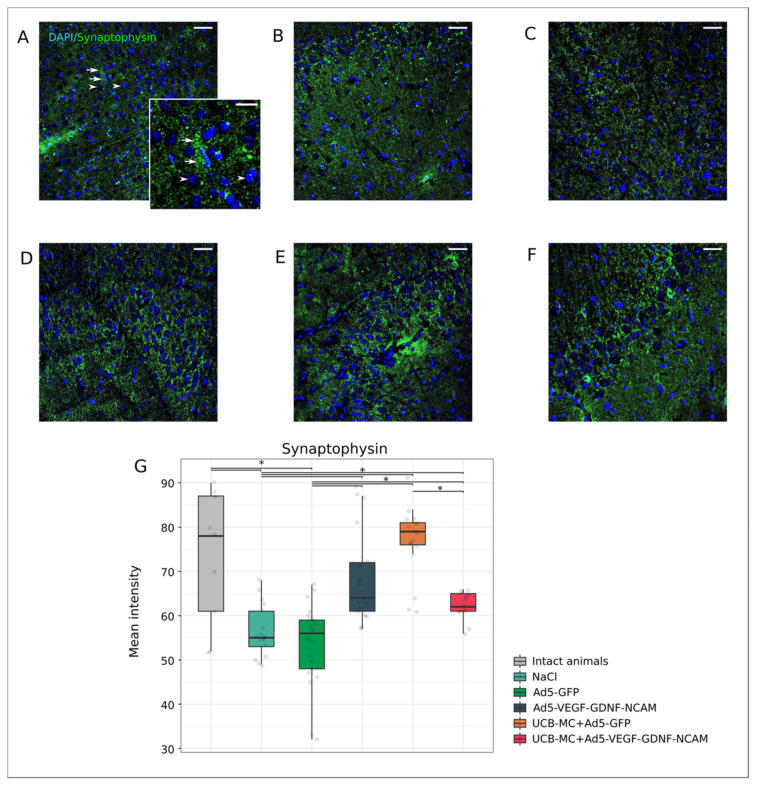
Immunoexpression of Synaptophysin in the rat brain cortex, 21 days after the middle cerebral artery occlusion. Immunofluorescent staining with antibody to synaptic vesicle membrane protein Synaptophysin in intact (**A**), control NaCl (**B**) and Ad5-GFP (**C**), and therapeutic UCB-MC+Ad5-GFP (**D**), Ad5-VEGF-GDNF-NCAM (**E**) and UCB-MC+Ad5-VEGF-GDNF-NCAM (**F**) groups. Arrows indicate Synaptophysin (green glow) in neurons. Arrowheads point to cell nuclei visualized using DAPI (blue glow). Scale bar in (**A**—**F**) = 50 µm; scale bar in the insert = 20 µm. (**G**)—comparative analysis of fluorescence intensity level of Synaptophysin in experimental groups; *—*p* < 0.05.

**Figure 9 ijms-21-06858-f009:**
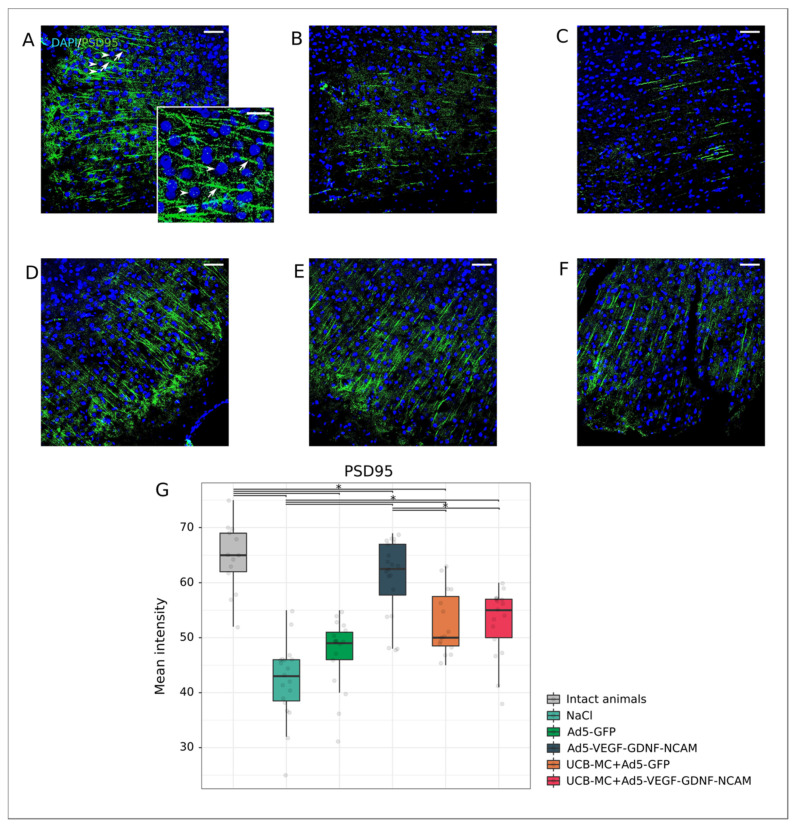
Immunoexpression of PSD95 in the rat brain cortex 21 days after the middle cerebral artery occlusion. Immunofluorescent staining with antibody to postsynaptic membrane protein PSD95 in intact (**A**), control NaCl (**B**) and Ad5-GFP (**C**), and therapeutic UCB-MC+Ad5-GFP (**D**), Ad5-VEGF-GDNF-NCAM (**E**) and UCB-MC+Ad5-VEGF-GDNF-NCAM (**F**) groups. Arrows indicate PSD95 (green glow) in neurons. Arrowheads point to cell nuclei visualized using DAPI (blue glow). Scale bar in (**A**–**F**) = 50 µm; scale bar in the insert = 20 µm. (**G**)—Comparative analysis of fluorescence intensity level of PSD95 in experimental groups; *—*p* < 0.05.

**Figure 10 ijms-21-06858-f010:**
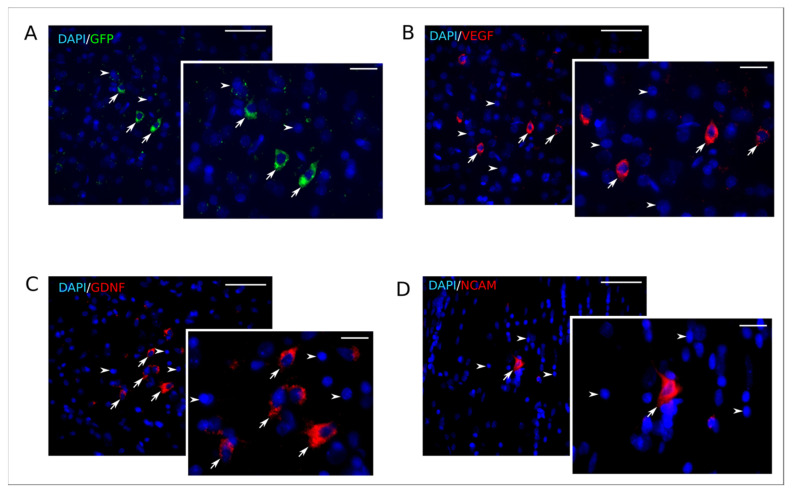
Expression of recombinant molecules in rat brain on 25th day after intrathecal injection of Ad5-GFP and Ad5-VEGF-GDNF-NCAM. (**A**)—Specific green glow of GFP in the transduced brain cells. (**B**)—Immune reaction with antibody to vascular endothelial growth factor (VEGF) in brain cells (red glow). (**C**)—Immune reaction with antibody to glial-derived neurotrophic factor (GDNF) (red glow). (**D**)—Immune reaction with antibody to neuronal cell adhesion molecule (NCAM) (red glow). Arrows point to the transduced brain cells. Arrowheads indicate cell nuclei visualized using DAPI (blue glow). Scale bar in (**A**–**D**) = 50 µm; scale bar in the inserts = 20 µm.

**Figure 11 ijms-21-06858-f011:**
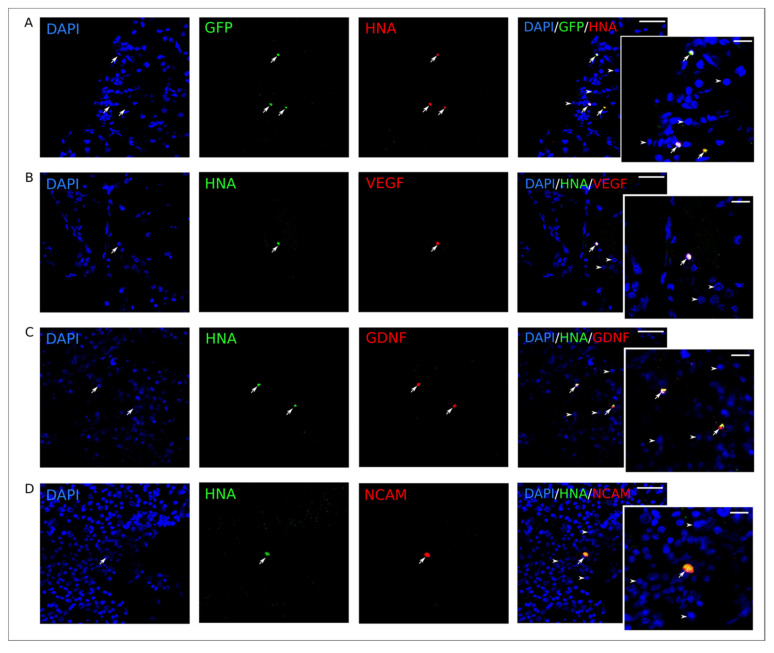
Expression of recombinant molecules in rat brain on the 24th day after intrathecal injection of UCB-MC+Ad5-GFP and UCB-MC+Ad-VEGF-GDNF-NCAM. UCB-MC+Ad-VEGF-GDNF-NCAM are visualized with antibody against the human nuclear antigen (HNA). Cell nuclei are counterstained with DAPI (blue glow). (**A**)—Specific green glow in UCB-MC+Ad5-GFP. (**B**)—Immune reaction with antibody against vascular endothelial growth factor (VEGF) (red glow). (**C**)—Immune response with antibody against glial-derived neurotrophic factor (GDNF) (red glow). (**D**)—Immune response with antibody against neuronal cell adhesion molecule (NCAM) (red glow). Arrows point to the UCB-MC. Arrowheads indicate cell nuclei visualized using DAPI (blue glow). Scale bar in (**A**–**D**) = 50 µm; scale bar in the inserts = 20 µm.

**Table 1 ijms-21-06858-t001:** Experimental groups of animals.

Groups	Preparation for Animals Treatment	Number of Animals
Control	20 µL 0.9% NaCl intrathecally injected 4 days before ischemic stroke modelling	5
Ad5-GFP	Ad5 carrying *gfp* in 20 µL of saline intrathecally injected 4 days before ischemic stroke modelling	6
Ad5-VEGF+Ad5-GDNF+Ad5-NCAM	Mixture of the three Ad5 carrying *vegf165*, *gdnf* and *ncam1* in 20 µL of saline intrathecally injected 4 days before ischemic stroke modelling	6
UCBC+Ad5-GFP	2 × 10^6^ UCBC transduced with Ad5 carrying *gfp* in 20 µL of saline intrathecally injected 3 days before ischemic stroke modelling	6
UCBC+Ad5-VEGF-GDNF-NCAM	2 × 10^6^ UCBC simultaneously transduced with three Ad5 carrying *vegf165*, *gdnf* and *ncam1* in 20 µL of saline intrathecally injected 3 days before ischemic stroke modelling	6

**Table 2 ijms-21-06858-t002:** Antibodies used in immunofluorescent analysis.

Antibody against:	Host	Dilution	Source
Caspase3	Rabbit	1:200	Abcam
Glial cell-derived neurotrophic factor (GDNF)	Rabbit	1:100	Santa Cruz
Glial fibrillary acidic protein (GFAP)	Mouse	1:200	Santa Cruz
Ionized calcium binding adaptor molecule 1 (Iba1)	Rabbit	1:150	Biocare Medical
Human Nuclear Antigen (HNA)	Mouse	1:150	Millipore
Heat shock protein 70 kDa (Hsp70)	Rabbit	1:200	Abcam
Oligodendrocyte transcription factor 2 (Olig2)	Rabbit	1:100	Santa Cruz
Neural cell adhesion molecule (NCAM)	Rabbit	1:100	Santa Cruz
Postsynaptic density protein 95 kDa (PSD95)	Rabbit	1:200	Abcam
Synaptophysin	Rabbit	1:100	Abcam
Vascular endothelial growth factor (VEGF)	Goat	1:300	Sigma
Rabbit IgG conjugated with Alexa 647	Donkey	1:200	Invitrogen
Mouse IgG conjugated with Alexa 488	Donkey	1:200	Invitrogen

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
