# Peer review of "Preventive Triple Gene Therapy Reduces the Negative Consequences of Ischemia-Induced Brain Injury after Modelling Stroke in a Rat"

_ijms, 2020, doi:10.3390/ijms21186858_

Round 1

Reviewer 1 Report

In the present study, authors have sought to investigate the effect of transplantation of engineered UCB-MC overexpressing recombinant VEGF, GDNF, and NCAM in a rat model of stroke. Authors found that, compared with controls, transplantation of engineered UCB-MC overexpressing recombinant VEGF, GDNF, and NCAM exerted not only cytoprotective effects but also promoted neurogenesis, while it suppressed gliogenesis. In addition, authors showed that, compared with controls, transplantation of engineered UCB-MC overexpressing these triple genes promoted functional recovery. I think this paper is well organized and interesting. However, I have raised several comments to improve the quality of this manuscript.

Major:

1. Results section: Please provide detailed illustration for each panel (e.g., A, B, C, D, E in Figure 2) in Figure 2, 3, 4, 5, 6, 7, 8 and 9.

2. Figure 2C and D: I think the coronal slice regions are a bit different between these panels. Please estimate histological findings using coronal brain sections of exact same regions.

3. Information about histological findings at the site of injured areas in Figure 3-11 is lacking. Are they obtained from within ischemic areas or around ischemic areas? It is better to provide the images together with schematic illustration so that one can understand the location of images easily.

4. Figure 3: It is better to provide the higher magnification of images so that one can understand the localization of HSP70+ cells.

5. Figure 5 and 6: I think it is usually impossible to count the number of Iba1+ cells and GFAP+ cells in vivo, while it is possible in vitro. I recommend that authors measure positive areas for Iba1 or GFAP (e.g., using NIH image software) and re-evaluate the results among groups.

6. Figure 10: Which populations of GFP+ cells co-express VEGF, GDNF, or NCAM?

7. Figure 11: It is likely that expression for VEGF, GDNF, or NCAM is only observed in HNA+ cells. Why endogenous brain cells other than transplanted cells did not express VEGF, GDNF, or NCAM? Also, why transplanted cells retained these expressions for a long time and did not deliver factors (VEGF, GDNF, or NCAM) to cells around them?

Minor:

1. Page 7, Line 138~141:“2.3. Immunofluorescent Study of Brain Comparative analysis・・・from intact animals.”can be removed into“methods section” because they are no description about results related to this.  

2. Results section: There are many points showing the data like this (AAA [BBB; CCC]) in Figure 2-9. AAA, mean? BBB, minimum? CCC, maximum? Please provide the meaning of each term.

3. Please provide full spell for all abbreviations (e.g., HNA).

Reviewer 2 Report

General comment.

This manuscript demonstrate the validity of the triple-gene (VEGF, GDNF, and NCAM) preventive measure by adenovirus or UCB-MCs that overexpress the three genes. While it is often difficult to predict stroke incident and hence cannot be a therapeutic in its current form, the effectiveness of the gene expression to the inhibition of stroke damage is worth reporting.

Minor: (all optional).

Consider MCAO instead of OMCA

It would be more convincing if ischemic damage was quantified by TTC staining.

Data in Fig 7 do not indicate the loss of myelination, while the result section says “by the destruction of oligodendrocytes and the subsequent demyelination of neural processes.”

2.3.3. Synaptic proteins section can be improved by synaptosome purification + western blotting. Immunohistochemistry alone is weak evidence because synaptic protein immunohistochemistry largely depends on the quality of fixation.

2.5 Distribution of migrated UCB-MC cells should be done.

Round 2

Reviewer 1 Report

Although authors improved original manuscript, they did not still response to all issues adequately. Please read comments carefully and revise the manuscript again.

  1. Results section for Figure 2, 3, 4, 5, 6, 7 ,8, and 9: The description for each panel is still lacking in text. As authors described in Figure 1, 10, and 11, the explanation for each panel (e.g, A, B, C・・・) is needed in text as well as Figure legends.

  1. Figure 5 and 6: Although some previous studies may evaluate GFAP or Iba1 expressions by counting cell numbers in vivo, I think it is NOT a right method. As you know, it is apparent that GFAP or Iba1 expressions are frequency detectable in cellular process (although some GFAP or Iba1 expressions are certainly observed in cytoplasm) in vivo. So, in general we cannot exactly evaluate GFAP or Iba1 expressions by counting cell numbers in vivo. Therefore, I strongly recommend that authors re-evaluated GFAP or Iba1 expressions by measuring positive areas for these markers.

  1. Figure 10: I just asked what percentage of GFP+ cells (e.g., 20%, 50%, 100%) continued to express VEGF, GDNF, or NCAM in vivo after transplantation.

Round 3

Reviewer 1 Report

Authors improved revised manuscript. However, as for Fig. 10, authors should provide the following data that I have requested.

Figure 10: It is very important to understand that what percentages of GFP+ transplanted cells continued to express VEGF, GDNF, or NCAM in vivo. So, please perform double immunohistochemistry for GFP and VEGF, GFP and GDNF, or GFP and NCAM. Then, analyze the percentage of VEGF+ cells of GFP+ cells, GDNF+ cells of GFP+ cells, or NCAM+ cells of GFP+ cells.

Round 4

Reviewer 1 Report

I agree with authors' comment. I think authors significantly improved their manuscript.